# Imaging inflammation using an activated macrophage probe with Slc18b1 as the activation-selective gating target

Sung-Jin Park [1], Beomsue Kim[1], Sejong Choi[2], Sivaraman Balasubramaniam[1], Sung-Chan Lee[1], Jung Yeol Lee[3], Heon Seok Kim[2], Jun-Young Kim[1], Jong-Jin Kim[1,4], Yong-An Lee[1], Nam-Young Kang[1,5], Jin-Soo Kim [2,6] & Young-Tae Chang [1,3,4]

Activated macrophages have the potential to be ideal targets for imaging inflammation. However, probe selectivity over non-activated macrophages and probe delivery to target tissue have been challenging. Here, we report a small molecule probe specific for activated macrophages, called CDg16, and demonstrate its application to visualizing inflammatory atherosclerotic plaques in vivo. Through a systematic transporter screen using a CRISPR activation library, we identify the orphan transporter Slc18b1/SLC18B1 as the gating target of CDg16.

[1] Laboratory of Bioimaging Probe Development, Singapore Bioimaging Consortium, Agency for Science, Technology and Research, Singapore 138667, Republic of Singapore. [2] Department of Chemistry, Seoul National University, Seoul 08826, Republic of Korea. [3] Department of Chemistry, Pohang University of Science and Technology (POSTECH), Pohang 37673, Republic of Korea. [4] Center for Self-assembly and Complexity, Institute for Basic Science (IBS), Pohang 37673, Republic of Korea. [5] New Drug Discovery Center, Daegu-Gyeongbuk Medical Innovation Foundation (DGMIF), Daegu 41061, Republic of Korea. [6] Center for Genome Engineering, Institute for Basic Science (IBS), Daejeon 34126, Republic of Korea. These authors contributed equally: Sung-Jin Park, Beomsue Kim, Sejong Choi. Correspondence and requests for materials should be addressed to J.-S.K. (email: jskim01@snu.ac.kr) or to Y.-T.C. (email: ytchang@postech.ac.kr)

Macrophages (Mφ) play many important roles in the immune responses of infected tissues through a polarized activation phase. Activated macrophages (Mφ*) are mainly classified as M1 (pro-inflammation) and M2 (anti-inflammation) macrophages, which can be induced by the in vitro treatment of lipopolysaccharide (LPS)/interferon-gamma (IFNγ) and interleukin-4 (IL-4)/IL-13, respectively[1]. Considering both M1 and M2 macrophages have important roles for the inflammatory processes of phagocytosis, antigen presentation, and scavenging activities (M1), as well as for the processes of wound-healing and tumor growth (M2), the targeted detection of both Mφ* has long been regarded as a direct approach for the diagnosis and prognosis of inflammatory diseases such as Alzheimer's dementia, hepatitis, atherosclerosis, and cancer[2–8].

Nonetheless, currently available imaging probes for live inflammation are mainly designed against indirect targets, such as adhesion molecules of endothelial cells in the inflamed area[6], metabolic targets of glucose consumption[7], and extracellular enzymes, including cathepsins[9] and matrix metalloproteinase (MMP)[10]. For example, LaRee1 and LaRee5 fluorescent probes were developed for imaging pulmonary inflammation using Foerster resonance energy transfer effect initiated by the membrane-bound MMP-12 enriched in the inflamed area[11]. PhagoGreen stained phagocytic macrophages in zebrafish[12]. The qABP probe labeled polyps in intestinal cancer by topical application with targeting cysteine cathepsins for the optical fluorescent imaging[13].

Although direct targeting of macrophages is a promising alternative approach, discrimination between non-activated (Mφ) and activated macrophages (Mφ*) is challenging. For such specific imaging of inflammation, a selective probe that only recognizes Mφ* would be ideal. Currently, a few targets, such as translocator protein (TSPO)[14] and folate receptor-β[15] are used for imaging Mφ* but low selectivity among macrophages and broad tissue expression of the proteins are limiting factors for whole body imaging. To overcome these limitations, we designed an unbiased screening of a fluorescent library using a polarized macrophage population, M1 macrophages, as a positive control and Mφ as a negative control. Here we report the successful development of a selective probe for Mφ*, CDg16. We demonstrate its application to imaging active inflammation in mice by direct targeting the accumulated Mφ* in the blood vessel wall of atherosclerosis[16], and uncover Slc18b1/SLC18B1 as a novel molecular target of the probe.

## Results

**Development of the activated macrophages probe, CDg16.** To construct the screening platform, Raw264.7 cells were used as Mφ and their activation by LPS (100 ng/mL) and IFNγ (20 ng/mL) was adopted to establish M1 macrophages[17]. The M1 polarized activation was confirmed by the generation of nitric oxide and the specific expression of M1 markers (inducible nitric oxide synthase (iNOS), CD38, and CD86), but not M2 marker (CD206), analyzed by immunocytochemistry (ICC) or flow cytometry (Supplementary Fig. 1). Over 8000 fluorescent library compounds were collected[18] and tested for Mφ and M1 macrophages side-by-side using a high-throughput imaging microscope (Supplementary Fig. 2). Compounds with higher fluorescence staining in M1 macrophages over Mφ were selected as the primary candidates. After a repeated screening, the probe with the best contrast and highest reproducibility was chosen as the final probe and dubbed CDg16 (Compound Designation green 16). CDg16 is a member of a novel acridine-based library (AD) (Fig. 1a, Supplementary Fig. 3, Supplementary Data 1 and Supplementary methods) and showed remarkable specificity and reliability for M1 macrophages

both in cell line and in primary mouse peritoneal macrophages (Fig. 1b). Specificity for M1 macrophages was confirmed by colocalization of the CDg16 stain and ICC of the M1 activated macrophage marker, CD86 (Supplementary Fig. 4). At the subcellular level, CDg16 localized to a population of lysosomal vesicles in M1 macrophages (Fig. 1c) and CDg16-positive vesicles appeared in polarizing cells from 8 h following activation of Raw264.7 cells with LPS and IFNγ (Fig. 1d). Interestingly, however, the CDg16-stained lysosomal vesicles were not merged to the low pH area of M1 macrophages (Supplementary Fig. 5a, white arrows showing the CDg16[bright]pHrodo[dim] vesicles). The independency of the CDg16 staining with low pH was further confirmed by the co-staining of CDg16 with the pHrodo-conjugated zymosan bioparticles to label low pH phagocytotic vesicles of M1 macrophages. CDg16 signals were not colocalized with the pHrodo-zymosan-derived fluorescent signals (Supplementary Fig. 5b).

To further examine the correlation between pH and the CDg16 staining, we compared CDg16 with a popular acridine-based pH-sensitive probe, acridine orange, which showed no specificity to M1 macrophages (Supplementary Fig. 6). For the comparison of chemical properties between the two probes, the calculated distribution coefficient (ClogD) and topological polar surface area (tPSA) values of acridine orange (AO) and CDg16 were calculated by ChemAxon (chemicalize program) for predicting and explaining the biodistribution of probes. Interestingly, although the ClogD values[19] of AO and CDg16 were similar (2.93 and 3.31 at pH 7.4 and 2.01 and 1.78 at pH 4.5, respectively), the tPSA value[20] was much lower in AO compared with CDg16 (19.4 versus 114.4). It suggests that CDg16 may be less (passively) permeable to the cells rather than AO, hence a unique mechanism such as specific transport may be involved in the CDg16 staining to Mφ* (Supplementary Fig. 6).

Notably, the low background staining of CDg16 enabled time-lapse imaging throughout the entire activation process without the need to wash the probe. CDg16 showed no apparent toxicity or disturbance to the macrophage activation process for 36 h (Fig. 1d and Supplementary Movie 1). Next, the universality of CDg16 was further examined by applying the probe to other types and origins of M1 macrophages. In comparison with their counterpart Mφ, consistently brighter staining pattern was observed in LPS/IFNγ-activated primary microglia from mouse brain (Supplementary Fig. 7a), M1 macrophages derived from human blood monocytes (Supplementary Fig. 7b), and M1 macrophages from the human macrophage cell line THP-1 (Supplementary Fig. 7c). To test if CDg16 can be systemically applied in vivo to whole animals, a topical acute inflammation animal model was induced by LPS injection into the paw areas of mice (Fig. 1e). After 1 h of intravenous (i.v.) injection of CDg16, LPS-injected (experimental) paws showed much higher numbers of CDg16-stained cells compared with phosphate-buffered saline (PBS)-injected (control) paws. The activation state of M1 macrophages was confirmed by colocalization of CDg16-stained cells (green) with CD86-positive cells (red) by immunohistochemistry (IHC) (Fig. 1f).

**Detecting atherosclerotic plaques using CDg16.** Atherosclerosis is a well-known inflammatory disease in humans and its progression is highly correlated with the population of Mφ* macrophages composed of mainly M1 and few M2 macrophages in arterial walls[16]. In light of this, we next tested CDg16 probe in the model of atherosclerosis in cell and animals. First, oxidized low-density lipoprotein (oxi-LDL) treated Raw264.7 Mφ were used as a model for atherosclerosis because oxi-LDL is one of the main risk factors for the accumulation of Mφ* at atherosclerotic blood

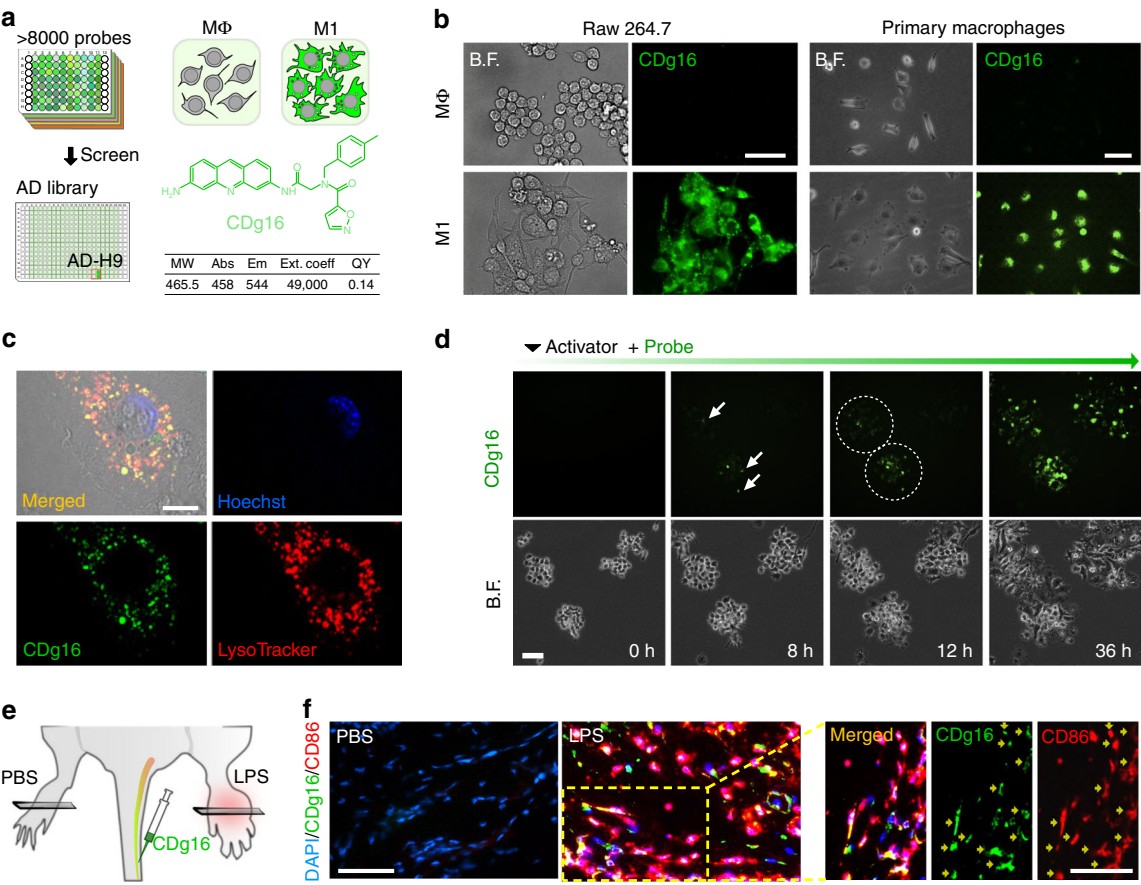

**Fig. 1** CDg16-stained activated macrophages. **a** The screening process and structure of the selected fluorescent probe, CDg16. Table shows the molecular/ fluorescence properties of CDg16. **b** Mφ (non-activated macrophages) and lipopolysaccharide (LPS) and interferon-gamma (IFNγ) activated M1 (classically activated macrophages) were used to examine the selectivity of CDg16. CDg16 stained only LPS and IFNγ treated Raw264.7 cells. LPS and IFNγ activated peritoneal macrophages (primary macrophages) were also stained by CDg16. **c** CDg16 signals colocalized with LysoTracker, which stains lysosomes. **d** With time tracking, CDg16 signals were found to appear 8 h after LPS and IFNγ treatment. Arrows indicate stained cells and dotted circles show the tracking of the CDg16-stained area. **e** Schematic of CDg16 application for the LPS-injected acute inflammation paw model with CDg16 tail vein injection. **f** CDg16 showed strong staining in the LPS-injected paw (LPS panel) and CDg16-positive cells colocalized with an activated macrophage marker (CD86 IHC, yellow arrows). The merged image was magnified from the inset yellow dotted square. Scale bars, 50 μm (**b**, **c**, **e**, **f**) and 10 μm (**d**). Data are representative of at least three independent experiments unless indicated otherwise

vessel wall and plaque areas, which directly induce the activation of macrophages[21] (Supplementary Fig. 8a). Along with the fact that oxi-LDL treatment preferentially induced the differentiation of Mφ to M1 macrophages (Supplementary Fig. 8b), the oxi-LDL-induced activated macrophages brightly stained by CDg16 compared with Mφ (Fig. 2a). Next, visualization of atherosclerotic plaques was directly tested in ApoE knockout (ApoE KO) mice (Fig. 2b), which were fed a western diet in order to stimulate plaque formation. Atherosclerotic plaques in ApoE KO mice extensively formed along the root of aorta arch (RAA), thoracic aorta (TA), and abdominal aorta (AA) from the heart (Figs. 2b, c and Supplementary Fig. 8c, d)[5,10]. As expected, after tail vein injection of CDg16, high fluorescence signals appeared in the severely atherosclerotic areas of RAA, TA, and the right brachiocephalic artery (RtB) of ApoE KO mice (yellow arrows in Fig. 2c, arrowheads in the CDg16 + ApoE group of Supplementary Fig 8c, d). The single-frame image with aortas of control mice clearly revealed that only CDg16-injected ApoE KO mice had high fluorescence signals in atherosclerotic plaques over the fluorescence levels of autofluorescent signals of Mφ* (Fig. 2d). Localization of CDg16 in M1 macrophages of atherosclerotic plaques was confirmed by the colocalization of the CDg16 signal with CD86 and iNOS (Fig. 2e and Supplementary Fig. 10),

whereas control aortas did not show any fluorescent signals except elastic laminar autofluorescence (Supplementary Fig. 9a). Very low CDg16 fluorescence from other organs, except fat pads, under the same optical imaging conditions (the exposure time and the binning) of the aorta observation, demonstrated that the injected probe preferentially accumulated in Mφ* in the plaque areas of atherosclerotic ApoE KO mice in vivo (Supplementary Fig. 9b). Moreover, it was clear that the strength of the fluorescence signal from fat pads was relatively low when directly compared with CDg16 fluorescence signals from plaques of the AA with pre-aorta fat in CDg16-injected ApoE KO mice (Supplementary Fig. 9c).

Next, we analyzed whether CDg16 labels M2 macrophages because M2 (alternative activated) macrophages also existed as a minor Mφ* populations in the atherosclerosis aorta (Supplementary Fig. 10)[16]. After confirming the in vitro differentiated M2 THP-1 macrophages by the expression of CD206 (Supplementary Fig. 11a, c), we found that both THP-1-derived M1 and M2 macrophages were strongly stained with CDg16 compared with non-activated control macrophages (Supplementary Fig. 11b). Flow cytometry analysis with live cells isolated from aorta tissue showed that the CD45+CD86+ (47.7%) or CD45+CD38+ (41.6%) M1 population in atherosclerosis specifically stained

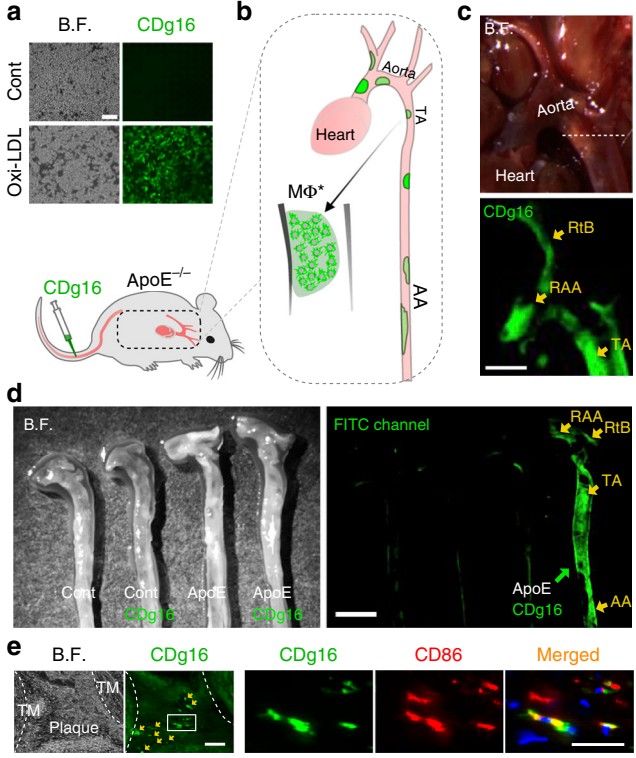

**Fig. 2** CDg16 application for detecting atherosclerotic plaques. **a** CDg16-stained Raw264.7 macrophages that were activated by oxidized low-density lipoprotein (oxi-LDL) treatment. **b** Schematic of CDg16 application for detecting atherosclerotic plaque areas by staining activated macrophages (Mφ*). **c** CDg16 was applied to control and ApoE knockout (KO) mice and CDg16 signals were observed using a fluorescent stereomicroscope. CDg16 only showed strong signals in CDg16-injected ApoE KO mice, specifically in plaque areas of the RAA, RtB, and TA. **d** After harvesting aortas from control and ApoE KO mice, CDg16 signals were compared under the FITC channel. Only CDg16-injected ApoE KO mice (ApoE CDg16) showed strong signals (yellow arrows). **e** Tissue sections along the dotted line in **c** were imaged for CDg16 signals and stained with CD86 antibody in order to detect activated macrophages. ApoE ApoE knockout, RAA the root of aorta arch, RtB right brachiocephalic artery, TA thoracic aorta, TM tunica intima. Scale bars, 100 μm (**a**), 1 mm (**c**), 2 mm (**d**), and 20 μm (**e**). Data are representative of at least three independent experiments unless indicated otherwise

with CDg16 compared with the other CD45⁻ cells (Supplementary Fig. 12b, d). Consistently with the staining of in vitro differentiated M2 macrophages, a few CD45+CD206+ M2 macrophages (2.1%) was also stained with CDg16 in atherosclerosis aorta tissues (Supplementary Fig. 12c). The specificity of CDg16 to Mφ* over other cell types was further confirmed by using in vivo atherosclerosis aorta cells (Supplementary Fig. 13a) and in vitro cell lines of endothelial, and smooth muscle origin (Supplementary Fig. 13b), as well as epithelial cell-derived human cancer cell lines (Supplementary Fig. 13c).

Finally, we examined the application of CDg16 to liver, another type of tissues, from control, ApoE KO and hepatitis mouse (Supplementary Fig. 14). Administration of CDg16 via tail vein discriminated M1 macrophages in the liver of ApoE KO and hepatitis to the control liver tissues of wild-type mouse, indicating that CDg16 can detect M1 macrophages regardless of tissue types.

**Slc18b1-mediated uptake of CDg16 in activated macrophages.** Next, we questioned how CDg16 labels Mφ* specifically. Since we discovered CDg16 via unbiased screening without any biomarker

information, it was necessary to narrow down the potential targets based on its staining characteristics. We observed two important phenomena: first, CDg16 fluorescence signals were completely removed after permeabilization, followed by fixation of stained M1 macrophages (Supplementary Fig. 15a). This suggests that CDg16 may solely reside in M1 macrophages rather than strongly bind to a biomolecular target. Second, the intracellular localization of CDg16 to sub-lysosomal vesicles was only observed in live, but not in dead M1 macrophages, implying an active transport process of live M1 macrophages may be involved (Supplementary Figs. 15b, 16). Endocytosis-mediated processes are a general means to uptake various substances from small molecules to complex macromolecules. To test the possibility of the involvement of endocytosis-mediated processes in the vesicular accumulation of CDg16, we used the drugs, cytochalasin D, LY294002, nystatin, filipin III, and phenylarsine, to inhibit endocytosis, macropinocytosis, micropinocytosis, clathrin-independent micropinocytosis, and micropinocytosis/phagocytosis, respectively. However, none of the inhibitors affected CDg16 accumulation in M1 macrophages, suggesting that CDg16 may enter the vesicles by another active mechanism (Supplementary Fig. 17).

We next focused on solute carrier (SLC) transporters, which import nutrients and xenobiotic molecules into live cells including phagocytic process[22,23]. Despite the importance of SLC transporters to a live organism, only a few members of SLCs have been extensively studied by their relevance to pharmacology and drug discovery[24]. Accordingly, there is no systematic tool currently available for screening SLC transporters, which comprise nearly 400 members[24]. We, therefore, attempted to create a novel systematic approach, SLC-CRISPRa (CRISPR activation), to screen SLC transporters for target identification of CDg16. Initially, the 380 protein-encoded SLC genes were selected from NCBI Gene (http://www.ncbi.nlm.nih.gov/gene) (Supplementary Data 2). By designing 10 single guide RNAs (sgRNAs) to the promoter region of each SLC gene, we successfully generated SLC-CRISPRa pools expressing one of the 3800 sgRNAs with dCas9-VPR[25] (Fig. 3a, Supplementary Fig. 18 and Supplementary Data 3). Next, the schematic screening process was verified with the two known fluorescent substrates, 4-Di-1-ASP and C1-BODIPY-C12, which are imported to intracellular spaces or fatty-acid rich vesicles of live cells by SLC22A23 and SLC27A2, respectively[26,27]. The expected SLC targets were successfully emerged from the six-round enriched population after gradually enriching the brightly stained population via fluorescence-activated cell sorting (FACS) (Supplementary Fig. 19). We then applied CDg16 to the SLC-CRISPRa system to identify SLC(s) that can selectively import the probe into vesicles. After six rounds of expansion of the top 3% of brightest populations from mother pools, the sorted CDg16bright population showed greater staining of CDg16 than the unsorted population (Fig. 3b and Supplementary Figs. 20, 21a). Through next-generation sequencing (NGS) analysis of the population in the sorted SLC-CRISPRa pools, three enriched sequences targeted to SLC18B1, SLC10A4, or SLC41A3, were shortlisted comprising 85.3% of the whole population (Fig. 3c). When the three SLC-sgRNA sequences were overexpressed individually, only SLC18B1-targeted sgRNA-transduced cells showed significantly enhanced CDg16 fluorescence (Fig. 3d). The correlation between SLC18B1 protein and CDg16 staining was confirmed through the colocalization of fluorescence signals between SLC18B1-mCherry and CDg16 in CDg16bright vesicles using the SLC18B1–mCherry fusion protein, suggesting that SLC18B1 indeed transports CDg16 into vesicles (Fig. 3e). Importantly, Slc18b1 KO via CRISPR/Cas9 in the M1 Raw264.7 macrophages resulted in reduced CDg16 fluorescence compared with levels in control M1 macrophages, indicating that mouse Slc18b1, the homolog of human SLC18B1,

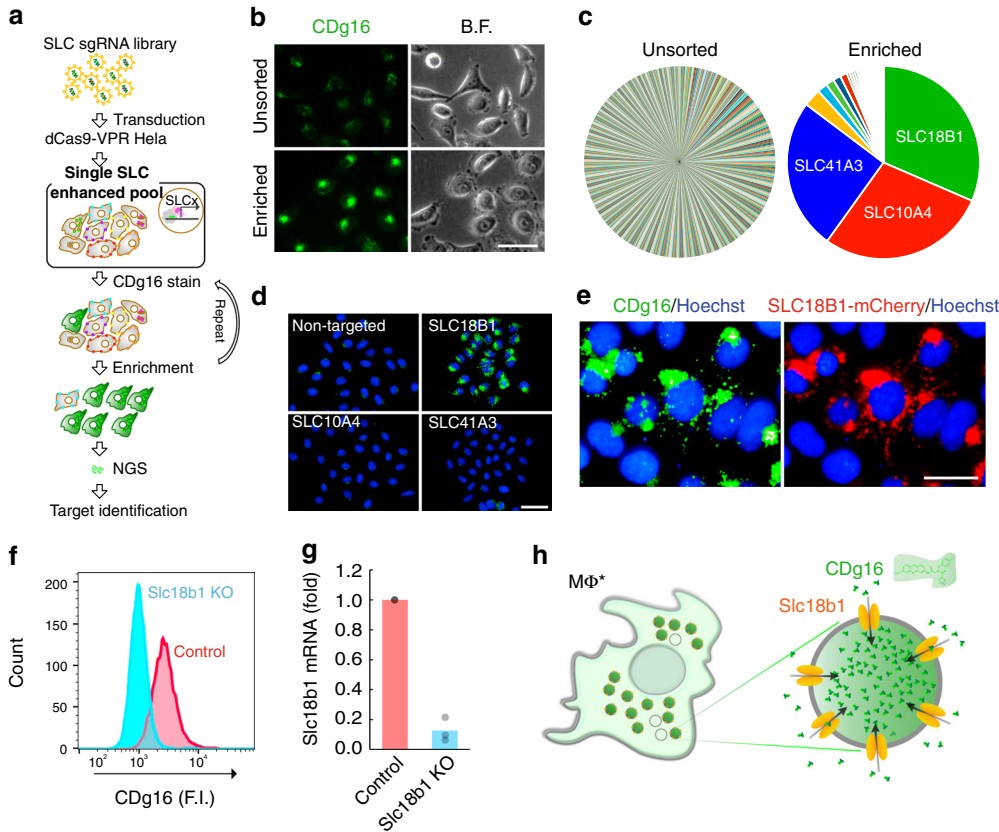

**Fig. 3** Slc18b1/SLC18B1-mediated uptake of CDg16. **a** Schematic of SLC-CRISPRa screening to detect SLC gene(s) for which CDg16 acts as a substrate. **b** CDg16 fluorescence and bright-field (B.F.) images of the unsorted SLC-CRISPRa HeLa cells (Unsorted) and the six-round enriched HeLa cells (Enriched). **c** NGS counts of the top 1000 highly enriched sgRNAs in the unsorted and the six-round enriched populations. The percentages of each sgRNA count to the total count are represented as a pie chart. The targeted genes for the three most enriched sgRNAs are indicated. **d** Fluorescence images of CDg16 (green) and Hoeschst33342 (blue) of the three independently generated SLC clones and non-targeted sgRNA. Fluorescence images were taken after 1-h incubation with CDg16 (200 nM) and Hoeschst33342 (1 μg/mL) in culture media. **e** Colocalization of the fluorescence signals of CDg16 (green) and of mCherry (red) from SLC18B1-mCherry transfected HeLa cells. Scale bars, 50 μm (**b**, **d**) or 20 μm (**e**). **f** The intracellular fluorescence intensity of control and Slc18b1-CRISPR knockout Raw264.7 M1 macrophages (Slc18b1 KO). CDg16 (250 nM) was incubated for 1 h to the LPS- (100 ng/mL) and IFNγ- (20 ng/mL) pre-treated M1. Flow cytometry was used to measure the intensity of CDg16 fluorescence. **g** Slc18b1 mRNA expression in control and Slc18b1 KO cells. **h** The proposed staining mechanism of CDg16. Mφ* activated macrophage, Slc18b1 solute carrier family 18 member b1

transport CDg16 in M1 macrophages (Fig. 3f, g and Supplementary Fig. 21b). We, therefore, suggest that Slc18b1 is the functional gating target of CDg16, and accumulates CDg16 in a type of lysosomal vesicles of M1 macrophages selectively (Fig. 3h). Moreover, like the overexpression of Slc18b1 in the LPS/IFNγ- or oxi-LDL-induced M1 macrophages, human SLC18B1 was highly expressed in both human M1 and M2 macrophages, supporting that SLC18B1 mediates the specific staining of CDg16 to Mφ* (Supplementary Fig. 22). As far as we know, CDg16 is the first substrate of the orphan transporter, SLC18B1.

Here, we present a novel optical imaging probe CDg16 for Mφ* by screening thousands of fluorescence library compounds and elucidated its staining mechanism as a selective entry through SLC18B1 transporter. CDg16 stains Mφ* selectively and successfully visualized the active inflammatory sites of atherosclerosis in animal model, overcoming the major hurdle for the targeted imaging of inflammation[28]. The development of a highly specific probe for activated macrophage in whole body by a simple i.v. injection will provide a unique diagnostic tool for inflammation-related diseases.

## Methods

**Preparation of CDg16 and synthesis of AD library.** CDg16 is the bio-fluorescence probe, which is discovered from the aminoacridine (AD)/AD chloroacetyl (ADCA) library. In total, 80 membered AD fluorescent library were designed based on its' natural fluorescence property. AD and ADCA were prepared from diamino-acridine (proflavine) core structure by the strategic expansion of its' biophore diversity.

AD library was synthesized on a solid support, which is well known as 2-chlorotrityl polystyrene resin. The amino group of AD is the labile tethering moiety, which enables us to diversify the biophore space. General loading method on polymer resin was used with pyridine in N,N-dimethylformamide/dimethyl sulfoxide and secondary amine group was introduced to go further amide products after chloroacetyl linker conjugation. ADCA is the modification of its' original version, AD, for the purpose of cellular incorporation. Detailed preparation methods and characterization data are described in the supplementary information (Supplementary Methods, Supplementary Fig. 23 to 33, Supplementary Data 1 and Supplementary Data 4).

**Cell culture for screening.** Mouse Raw264.7 macrophage cell line (ATCC® TIB-71™) was used for screening. Raw264.7 cells were cultured in a culture dish in high-glucose Dulbecco's modified Eagle's medium supplemented with 10% fetal bovine serum, 100 U/mL penicillin, 100 μg/mL streptomycin (Life Technologies). For making activated macrophages (Mφ*), 100 ng/mL LPS and 20 ng/mL IFNγ were treated in Raw264.7 cells for 24–48 h. Only the activated Raw264.7 cells showed the activation morphology of flattened spread cells were used for experiments, i.e., screening and intracellular localization.

**Activation of other macrophages.** To activate other macrophage cell lines or primary cells, LPS (100 ng/mL, Sigma-Aldrich) and IFNγ (20 ng/mL, Life Technologies) were treated for 24 h at 37 °C. For primary cell tests, mouse peritoneal macrophages from peritoneal cavity were isolated and collected macrophages were

activated by LPS and IFNγ. Mouse microglia was isolated from confluent glial cultures, which obtained by 2–3 weeks cultures of neonatal cortices. Human monocytes were collected from human peripheral blood by using Percoll gradient protocol. Human monocytes were differentiated to human macrophages with 50 nM phorbol-12-myristate-13-acetate, and the human macrophages were further activated by LPS and IFNγ to produce human activated macrophages. Human activated macrophages were stained with 500 nM CDg16 for 1 h at 37 °C.

**Screening**. For high-throughput screening, control and activated Raw264.7 cells plated in 384-well microplates were incubated with a probe at a concentration of 1 μM in duplicate. After 1 h, fluorescence and bright-field images were taken by using an ImageXpress^MICRO imaging system (Molecular Devices). From the primary screening with over 8000 fluorescence compounds, 14 fluorescent compounds stained activated Raw264.7 (activated macrophage) cells with stronger intensity than non-activated Raw264.7 cells. From a secondary and a tertiary screening, we narrowed down the candidates to one acridine chromophore motif probes for the further study.

**Time-tracking observation for activating macrophages**. Raw264.7 cells were plated on the cell culture plate and treated simultaneously with LPS (100 ng/mL), IFNγ (20 ng/mL), and CDg16 (1 μM). The several fixed positions of macrophages were continuously observed from pre-activation to post-activation, every 30 min for a total 36 h under the bright field and the green fluorescent protein channel. All observations were performed by the BioStation IM-Q time-lapse imaging system (Nikon).

**Animal experiment**. All animal experimental procedures were performed in accordance with a protocol approved by the Institutional Animal Care and Use Committee for Biological Resource Center at A*STAR, Singapore (IACUC #151032 and #151033). ApoE KO mice (apolipoprotein E-deficient mice, ApoE−/− (The Jackson laboratory)) fed a western diet were used for the atherosclerosis model. CDg16 (500 μM, 200 μL per 20 g mouse) was injected via tail vein for control and ApoE KO mice, and CDg16 signals of aorta area were observed by the customized fluorescent stereomicroscope (Leica Microsystems). After the fluorescent imaging, the aorta was enucleated and evaluated by IHC.

**Immunofluorescence staining**. The aorta and paw samples were enucleated and immediately frozen for the cryo-sections. The samples were sectioned by the cryostat (Leica CM1950) with 10 μm thickness and mounted on the poly-L-lysine-coated slides. The sectioned samples were fixed in 4% paraformaldehyde (PFA) for 15 min for IHC. The cell culture samples were also fixed in 4% PFA for 15 min for ICC. After washing the sectioned and cell culture samples with PBS, the samples were treated with 1% bovine serum albumin (30 min) for removing nonspecific binding. Rat anti-CD86 antibody (dilution 1:100, BD Pharmigen, 553689) was incubated overnight at 4 °C for staining activated macrophages. For secondary antibody staining, Alexa 647-conjugated goat anti-rat IgG (dilution 1:500, ThermoFisher Scientific, A-21247) was used. All images were taken by Eclipse Ti-E Microscopy (Nikon).

**sgRNA library design**. The targeted 380 human SLC genes having "SLC" in their official gene name were selected through the NCBI database (https://www.ncbi.nlm.nih.gov/gene) (Supplementary Data 1). The protospacer adjacent motif sequence containing sgRNA sites for each SLC gene was selected within its promoter region up to the 400 bp, resulting 3800 sgRNA-targeted sequence (Supplementary Fig. 18b and Supplementary Data 2).

**Lentiviral production**. HEK293T cells ($5 \times 10^6$ cells) (ATCC® CRL-3216™) were seeded on a 100-mm dish a day before transfection. Cells are transfected with the lentiviral library plasmids (15 μg) and the three virus packaging plasmids (9 μg of pMLDg.pRRE, 6 μg of pRSV-Rev, and 3 μg of pCMV-VSV-G) using Lipofectamine 2000 (Invitrogen). The lentiviral particles were harvested at 48 h after transfection and filtered using a 0.45-μm filter.

**Generation of SLC-CRISPRa pools**. For enhancing sufficient levels of gene transcription, we selected dCas9-VPR, which fused with the three different types of cis-acting transcription activation domain (Supplementary Fig. 18a)[25]. After confirming that the dCas9-VPR can activate the two different target genes, IL1RN and SLC28A2, with separate three sgRNAs, respectively, we generated a stable cell line expressing dCas9-VPR selected by G418 (500 μg/mL, Invitrogen) to the transfected Hela cells (ATCC® CCL-2™). The 3800 sgRNA libraries were stably overexpressed by infection of the lentiviral particles into the dCas9-VPR Hela to generate SLC-CRISPRa pools (Supplementary Fig. 18c). The sgRNA expressing cells were selectively expanded by incubating the cells with puromycin (2 μg/mL). To maintain the diversity of the pools, $>1.52 \times 10^6$ cells were plated for subsequent culture and used for the initial sorting with a fluorescent probe (Supplementary Fig. 18c).

**Fluorescent probe staining to live cells**. 4-DI-1-ASP and C1,C12-BODIPY were purchased from Sigma-Aldrich. SLC-CRISPRa pools or enriched cells were treated with 1 μM of each probe for 1 h. Hoechst33342 (1 μg/mL) was co-treated for labeling all nuclei of live cells if necessary.

N-terminal mCherry fusion construct of SLC18B1 (NM_052831.2) overexpression vector was purchased from GeneCopoeia (EX-T3513-M55). After 2 days of transient transfection of the plasmid DNA using Lipofectamine 3000 (Invitrogen), the SLC18B1-mCherry transfected Hela were stained with Hoechst33342 (1 μg/mL) and CDg16 (200 nM) for 30 min to observe the colocalization of the stained vesicle in live cells. The fluorescence images were obtained by using Observer Z.1 inverted microscope (Zeiss) or Operetta (Perkin Elmer).

**FACS**. Healthy SLC-CRISPRa Hela pools were detached by using Accutase (ThermoFisher Scientific) to minimize damages on their surface. A fluorescence probe (1 μM) was added to the pools under culture media for 30 min. Live singlets of the cell populations were sorted with the gating of 3% bright population of the fluorescent intensity by using MoFlo XDP cell sorter (Beckman Coulter). After sorting, the cells were cultured for a week to expand the sorted population before next-round of FACS. The sorting procedure was repeated for at least six rounds to ensure all the populations were highly enriched. The FACS data were analyzed by using FlowJo 10.2 software (TreeStar).

**Analysis of integrated sgRNA**. Genomic DNAs of the SLC-CRISPRa Hela pools and the sorted cell populations were isolated by using Purelink Genomic DNA Mini Kit (Invitrogen). The sgRNA containing region was amplified with primers F—5′-TCTTGTGGAAAGGACGAAACACCG-3′ and R—5′-TCTAC-TATTCTTTCCCCTGCACTGT-3′ with 10 μg of genomic DNA and sequenced using an Illumina HiSeq 4000 for 100× coverage. After sequencing, reads were aligned to the sgRNA library and counted. Those sgRNA counts are analyzed by MAGeCK (Model-based Analysis of Genome-wide CRISPR-Cas9 Knockout) version 0.5.5 according to the instructions[29].

**Generation of single SLC gene activated Hela**. The HeLa cells, which express dCas9-VPR stably, and lentiviral particles, which express a specific sgRNA, are infected. After confirming target gene overexpression by real-time qPCR compared with control, the single SLC gene activated Hela cells were used for subsequent fluorescence staining.

**CRISPR KO experiment**. Cas9 expressed Raw264.7 was generated by Blasticidin S selection, followed by the transduction of the cells with lentiCas9-Blast lentiviral particles (a gift from Feng Zhang, Addgene plasmid #52962). To generate SLC18B1 KO clones, Cas9 expressed Raw264.7 seeded on 12-well plate ($3 \times 10^4$ cells per well) was infected with SIGMA LentiCRISPR Slc18b1 (Target ID, MM0000620497; Target sequence, 5′-AGCGGCGAAGAAATGGCGTAGG-3′, Sigma). After 2 days incubation, the infected cells were selected with Puromycin (5 μg/mL) for 1 week. Survived cells were expanded and KO was checked by real-time polymerase chain reaction (RT-PCR) to the Slc18b1 expression.

**RT-PCR**. The total RNA was isolated using RNeasy Mini Kit (QIAGEN Inc.) from cultured cells according to the manufacturer's instruction. One-step quantitative RT-PCR was performed on a StepOne™ Real-Time PCR system using a Power SYBR® Green RNA-to-CT™ 1-Step Kit (Applied Biosystem). The relative mRNA levels of the genes were normalized to that of β-actin. The primer sequences for checking Slc18b1 KO examination were: F—5′-AAGAAGGGAGCCAGCAACAC CATG-3′ and R—5′-CAAACAGCGGCGAAGAAATGGCG-3′.

**Reporting summary**. Further information on experimental design is available in the Nature Research Reporting Summary linked to this article.

## Data availability

Any supplementary information and videos are available in the online version of the paper. The deep sequencing data that support the findings of this study have been uploaded to the NCBI Sequence Read Archive under Bioproject accession code PRJNA516962. All other data are available from the authors upon request.

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

## Acknowledgements
We thank Ms. Wut-Hmone Phue and Dr. Samira Husen Alamudi for technical assistance. We also appreciate Dr. Philip Lee who kindly provided hepatitis mouse model. This study was supported by an intramural funding program of the Joint Council Office (JCO) Career Development Award (CDA) and A*STAR (Agency for Science, Technology and Research, Singapore) (15302FG148), and of the Institute for Basic Science (IBS-R021-D1 to J.-S.K. and IBS-R007-A1 to Y.-T.C.).

## Author contributions
S.B., S.-C.L. and J-Y.L. synthesized library and characterized CDg16 probe. S.-J.P. and N.-Y.K. did screening for activated macrophage staining probe. S.-J.P., B.K., J-Y.K., Y.-A.L. and N.-Y.K. performed cell culture experiments. S.-J.P., J.-J.K. and Y.-A.L. did animal experiments. S.C., H.S.K. and J-S.K. developed CRSPRa platform for discovering CDg16 mechanism study. S.-J.P. B.K. and J.-Y.K. evaluated the putative target mechanism. S.-J.P., B.K., J-S.K. and Y.-T.C. reviewed, analyzed, and interpreted the data. S.-J.P., B.K. and Y.-T.C. wrote the paper. All authors discussed the results and commented on the manuscript.

## Additional information

**Competing interests:** S.-J.P., B.K., S.B., S.-C.L. and Y.-T.C. are the inventors of CDg16 for which a patent has been filed. The remaining authors declare no competing interests.

