## [Peer Review File · Nature Communications]

Reviewers' comments:

Reviewer #1 (Remarks to the Author):

The manuscript by Park et al. describes a small molecule fluorophore that stains pro-inflammatory monocyte-macrophage cells and facilitates ex vivo visualization of atherosclerotic plaques. The authors attribute the preferential accumulation of the probe to the recognition of Slc18b1 upon an extensive screening of transporters. The manuscript is well written and covering work of remarkable breadth, spanning from chemical synthesis of the fluorophore to its characterization in cells, tissues and in vivo models as well as the characterization and identification of putative biomolecular targets. The preparation of optical probes for visualization of atherosclerotic plaques is timely, with only a handful of successful examples in the literature. Overall, I think the manuscript can be considered for publication in Nature Communications provided that several aspects are improved and clarified.

1. I think the term 'activated macrophage' is perhaps oversimplified and it may be misleading. Macrophages can be classified as classically or alternatively-activated macrophages and display pro-inflammatory (what the authors call here in this manuscript activated) or anti-inflammatory phenotypes, respectively. This point and what the authors consider 'activated macrophages' should be clarified in the introduction. Likewise, the authors should also assess the behaviour of CDg16 in alternatively-activated macrophages (e.g. IL-4 stimulation), together with validation with CD206 or arginase as conventional anti-inflammatory markers. The pro-inflammatory phenotype was confirmed by CD86 staining but the authors should also perform functional assays (e.g. NO production) that would clarify the characteristics of the CDg16-stained population.

2. The probe CDg16 seems to localize in lysosomal compartments, which may be more acidic than the rest of the cell. The structure of CDg16 contains pH-sensitive groups but is the fluorescence of CDg16 brighter in acidic compartments? The authors could clarify this point and perhaps compare CDg16 to pHrodo as a pH-sensitive dye for staining phagocytic macrophages. Fig 1 may include the basic spectral properties of the probe (wavelengths of excitation and emission, quantum yield). The probe emits in the green region and macrophages often show green autofluorescence, did the authors need any special consideration to this point during the experiments?

3. The observation that CDg16 stains oxLDL-treated macrophages is interesting. Did the authors assess the levels of CD86, CD206 or the Slc18b1 transporter in these cells?

4. Notably, the authors used CDg16 in vivo in control and ApoE -/- mice. The results indicate preferential staining in aortas from ApoE-/- but only in a qualitative manner and based on ex vivo tissue images. Are the CDg16-stained cells pro-inflammatory macrophages? Further characterization (including quantitative data) of in vivo-stained cells is needed, including additional markers and flow cytometry. The authors also should include in vitro comparative analysis of CDg16 between macrophages and other cells (e.g. endothelial, smooth muscle) that could be found in the aorta tissue.

5. Finally, the authors performed a comprehensive screening using SLC-CRISPR to identify potential partners for CDg16. Interestingly, the authors identified Slc18b1 as a potential transporter of CDg16 in cells. These experiments however are performed in HeLa cells hence it is unclear whether this mechanism correlates to the preferential staining of pro-inflammatory macrophages in vitro and in vivo. This is an important point that should be clarified.

Minor:

-The introduction should be expanded clarifying the above-mentioned comments and also including examples of other fluorescent probes for imaging pro- or anti-inflammatory macrophages (e.g. cathepsins from Bogoyo et al, phagocytosis probes from Vendrell et al, MMP-12 from Schultz et al.).

-Copies of NMR and HRMS spectra of CDg16 should be added to the Supplementary Information.

Reviewer #2 (Remarks to the Author):

Park, SJ et al. "Imaging inflammation using an activated macrophage probe with Slc18b1 as the selective gating target" In this manuscript, the authors describe a library selection of a new probe based on an acridine orange parent structure for activated macrophages. Furthermore, the authors performed a biological selection to determine which transporter is responsible for the lead compound's import into activated macrophages. This is a well written manuscript with just a few grammatical errors. The search for a specific M1 polarization probe is important and a worthwhile goal. The authors performed in vitro and ex vivo fluorescence imaging in cultured cells and excised tissues of mice with limited correlative histologic confirmation of cell type or phenotype. This flaw and others listed below reduce the quality of this manuscript:

1. No ClogD or PSA values for acridine orange or for CDg16 are presented for comparison. These are crucial for predicting and explaining probe biodistribution in fatty tissues and in macrophages accumulating lipid as well as in systemic clearance and CNS uptake.
2. Acridine orange does not seem to have been used as a comparative control. It would appear, for example, that CDg16 simply accumulates in cells with high densities of acidic or lipid dense lysosomes (like A.O.). Epithelial cancer cells were not included in uptake studies and these contain many similar vesicles.
3. This reviewer did not see any attempt to perform blocking studies to ascertain specificity for cells. In the ApoE^{-/-} model, arteries contain plaques rich in lipid pools, which may non-specifically solubilize this particular probe and if so, probe would be extracted out during routine paraffin embedded histology methods. The photos provided are macroscopic and would not differentiate between fat solubilization and cell uptake. The use of unfixed, frozen sections would be very helpful in delineating total probe distribution.
4. Kupffer cells in liver (and macrophages in lung) in an ApoE^{-/-} mouse would be highly inflamed and no data were presented in these tissues as the identity of the livers and lungs in the supplementary data is unclear (control or inflamed?). Liver is dark and may easily quench visible green fluorescence, limiting the applicability of this probe.
5. Please change your designation from "activated macrophages" to "reactive M1 phenotype macrophages", if this is what you mean.
6. M2 macrophages are also reactive and have just as many acidic lysosomes and should also express the Slc18b1 transporter as many tissue types do. It stands to reason that M1 probe trapping may be related to ROS ionization, rather than lysosome sequestration. The anthracene ring should be quite susceptible if not the oxazole. No experiments were presented showing M2 polarized macrophages and probe uptake. Histologic confirmation was so limited, readers cannot conclude whether M1 specificity is, in fact, a robust phenomenon. CD86 is expressed by several antigen presenting cell types. No additional markers were used of any kind such as CD68, CD206, CD163 or even iNOS. With fluorescence imaging, multi-channel marker use is encouraged for clarity.

While the search for an M1 specific probe continues, it is the opinion of this reviewer that the work presented in this manuscript lacks the experimental rigor needed to support the conclusions.

Response to Reviewers

Reviewers' comments:

Reviewer #1 (Remarks to the Author):

The manuscript by Park et al. describes a small molecule fluorophore that stains pro-inflammatory monocyte-macrophage cells and facilitates ex vivo visualization of atherosclerotic plaques. The authors attribute the preferential accumulation of the probe to the recognition of Slc18b1 upon an extensive screening of transporters. The manuscript is well written and covering work of remarkable breadth, spanning from chemical synthesis of the fluorophore to its characterization in cells, tissues and in vivo models as well as the characterization and identification of putative biomolecular targets. The preparation of optical probes for visualization of atherosclerotic plaques is timely, with only a handful of successful examples in the literature. Overall, I think the manuscript can be considered for publication in Nature Communications provided that several aspects are improved and clarified.

We appreciate the encouraging comments from the reviewer.

Q1. The probe CDg16 seems to localize in lysosomal compartments, which may be more acidic than the rest of the cell. The structure of CDg16 contains pH-sensitive groups but is the fluorescence of CDg16 brighter in acidic compartments? The authors could clarify this point and perhaps compare CDg16 to pHrodo as a pH-sensitive dye for staining phagocytic macrophages.

We appreciated the valuable comments of the reviewer.

Following the reviewer's suggestion, we have compared CDg16 with pHrodo Red AM in activated Raw264.7 macrophages and found CDg16-stained vesicles were independent with the pHrodo Red AM-stained vesicles (Supplementary Figure 5a, white arrows showing the CDg16^{bright}pHrodo^{dim} vesicles). We also examined the pHrodo-conjugated zymosan bioparticles to label low pH phagocytotic vesicles and found minimum overlay between CDg16 and zymosan-pHrodo derived fluorescence signal in activated Raw264.7 cells (Supplementary Figure 5b). These results indicate that the behavior of CDg16 staining is independent to pH in the M1 macrophages.

Q2. Fig 1 may include the basic spectral properties of the probe (wavelengths of excitation and emission, quantum yield).

We added all the basic spectral properties of CDg16 into Figure 1a following the advice.

Q3. The probe emits in the green region and macrophages often show green autofluorescence, did the authors need any special consideration to this point during the experiments?

All experiments have been performed with proper negative control to assure the levels of green fluorescence are not from autofluorescence of macrophage. The cutoff levels of background fluorescence in the figures were much higher than the autofluorescence signals from non-stained macrophages. To ensure the results, we showed all the four group of aortas, i.e. wild-type alone, wild-type/CDg16-injected, ApoE^{-/-} alone, ApoE^{-/-}/CDg16-injected, into a single picture in Figure 2d and Supplementary Figure 8. Moreover, the histogram data of non-stained and CDg16 stained aorta cells by flow cytometer experiments clearly showed that the autofluorescence signal from non-stained aorta cells did not reach the levels of CDg16 fluorescence (Supplementary Figure 13a).

Q4. The observation that CDg16 stains oxLDL-treated macrophages is interesting. Did the authors assess the levels of CD86, CD206 or the Slc18b1 transporter in these cells?

We thank the reviewer for raising the interesting question for the oxLDL-treated macrophages. To confirm the activation and the type of oxLDL-treated macrophages, we analyzed the levels of CD86, CD206, and Slc18b1 by RT-PCR (Supplementary Figure 8b). As expected, oxLDL treatment stimulated CD86 expression, but not CD206, with a similar trend with the LPS/IFN- γ -induced M1 activation of macrophage (Supplementary Figure 8b). It indicates that oxLDL preferentially activates macrophages to M1 with the enhanced expression of the target molecule of CDg16, Slc18b1 (Supplementary Figure 21a).

Q5. Notably, the authors used CDg16 in vivo in control and ApoE^{-/-} mice. The results indicate preferential staining in aortas from ApoE^{-/-} but only in a qualitative manner and based on ex vivo tissue images. Are the CDg16-stained cells pro-inflammatory macrophages? Further characterization (including quantitative data) of in vivo-stained cells is needed, including additional markers and flow cytometry. The authors also should include in vitro comparative analysis of CDg16 between macrophages and other cells (e.g. endothelial, smooth muscle) that could be found in the aorta tissue.

As suggested by the reviewer, we performed the flow cytometry analysis with cells isolated from the aorta tissue of ApoE^{-/-} using the anti-CD38 and anti-CD86 antibodies for M1 and the anti-CD206 antibody for M2 macrophages labeling. We observed that the activated macrophages in the ApoE^{-/-} aorta are mainly M1 macrophages (47.7%), with very small populations of M2 macrophages (2.1%) (Supplementary Figure 12).

It is noteworthy that all the CD45⁻ populations (non-leukocytes) were negatively labeled with CDg16, which confirms CDg16 labeled only activated macrophage, but not the other cell types composing of the atherosclerosis aorta (Supplementary Figure 12). Further *in vitro* test with endothelial and smooth muscle cell lines showed clear negative staining by CDg16 (Supplementary Figure 13a).

Q6. I think the term ‘activated macrophage’ is perhaps oversimplified and it may be misleading. Macrophages can be classified as classically or alternatively-activated macrophages and display pro-inflammatory (what the authors call here in this manuscript activated) or anti-inflammatory phenotypes, respectively. This point and what the authors consider ‘activated macrophages’ should be clarified in the introduction. Likewise, the authors should also assess the behaviour of CDg16 in alternatively-activated macrophages (e.g. IL-4 stimulation), together with validation with CD206 or arginase as conventional anti-inflammatory markers. The pro-inflammatory phenotype was confirmed by CD86 staining but the authors should also perform functional assays (e.g. NO production) that would clarify the characteristics of the CDg16-stained population.

We thank for the reviewer’s comments. As the reviewer pointed out, LPS/IFN γ -activated Raw264.7 are regarded as M1 (classically-activated) macrophages and usually produced large amounts of nitric oxide (NO). We confirmed the high NO production from the activated Raw264.7 cells in 24 hours of activation by LPS/IFN- γ , the same condition used for screening (Supplementary Figure 1c). Additionally, we further confirmed its activation by flow cytometry analysis using CD38 in addition to CD86 (Supplementary Figure 1b) and also by immunofluorescence labeling using CD86 and iNOS antibodies (Supplementary Figure 1a). While all three M1 markers were remarkably expressed on LPS/IFN- γ -activated Raw264.7 macrophages compared to control cells, M2 marker CD206 showed no increase (Supplementary Figure 1).

Following the reviewer’s advice, we prepared M2 (alternatively activated) macrophages by IL-4/-13 stimulation. THP-1 monocytes were used to differentiate both to M1 and M2 macrophages with confirming its proper differentiation with CD86 and CD206 labeling each (Supplementary Figure 11a). Interestingly, M2 macrophages were also labeled by CDg16, suggesting that both M1 and M2 can be labeled by CDg16 (Supplementary Figure 11b, c). So, we may keep the target of CDg16 as originally designated as activated macrophages.

As recommended by the reviewer, we included the classification of M1 and M2 in the introduction (Line 34-42). While the distinguishment of M1 and M2 is another interesting topic, understanding the common part of the development is also important. As a result of our work, with the reviewer’s suggestion, we found the common basis of M1 and M2 in terms of the common target of CDg16, as of SLC18B1 expression. The discovered facts and discussions are described in the later part of the paper (Line 146-166, 213-221).

Q7. Finally, the authors performed a comprehensive screening using SLC-CRISPR to identify potential partners for CDg16. Interestingly, the authors identified Slc18b1 as a potential transporter of CDg16 in cells. These experiments however are performed in HeLa cells hence it is unclear whether this mechanism correlates to the preferential staining of pro-inflammatory macrophages in vitro and in vivo. This is an important point that should be clarified.

Thanks for the reviewer's point. We agreed to the reviewer's point that the potential transporter of CDg16 was screened by HeLa cells and need to be clarified in the activated macrophages. To confirm the result in the SLC-CRISPRa data, Slc18b1 was knockout from Raw264.7 macrophages to show that Slc18b1-knockout leads to the reduction of CDg16 staining of the M1 activated macrophage in Figure 3f. We revised the main text to emphasize the reviewer point in the Line 213 to 216 as followed: "Importantly, Slc18b1 knockout (KO) via CRISPR/Cas9 in the M1 Raw264.7 macrophages resulted in reduced CDg16 fluorescence compared to levels in control M1 macrophages, indicating that mouse Slc18b1, the homolog of human SLC18B1, transport CDg16 in M1 macrophages (Fig. 3f,g)."

Minor:

-The introduction should be expanded clarifying the above-mentioned comments and also including examples of other fluorescent probes for imaging pro- or anti-inflammatory macrophages (e.g. cathepsins from Bogoyo et al, phagocytosis probes from Vendrell et al, MMP-12 from Schultz et al.).

We have revised the introduction of the manuscript containing the previously reported probes for imaging pro- or anti-inflammatory macrophages as below.

Line 34 to 42

"Activated macrophages (M ϕ *) are mainly classified as M1 (pro-inflammation) and M2 (anti-inflammation) macrophages, which can be induced by the *in vitro* treatment of lipopolysaccharide (LPS)/interferon- γ (INF- γ) and interleukin-4 (IL4)/IL-13, respectively¹. Considering both M1 and M2 macrophages have important roles for the inflammatory processes of phagocytosis, antigen presentation and scavenging activities (M1) as well as for the processes of wound-healing and tumor growth (M2), the targeted detection of both M ϕ * has long been regarded as a direct approach for the diagnosis and prognosis of inflammatory diseases such as Alzheimer's dementia, hepatitis, atherosclerosis, and cancer²⁻⁸."

Line 46 to 51

"For example, LaRee1 and LaRee5 fluorescent probes were developed for imaging pulmonary inflammation using FRET effect initiated by the membrane-bound MMP-12 enriched in the inflamed area¹¹. PhagoGreen stained phagocytic macrophages in zebrafish¹². The qABP probe labeled polyps in intestinal cancer by topical application with targeting cysteine cathepsins for the optical fluorescent imaging¹³."

(Dr. Schultz et al., Dr. Vendrell et al., and Bogoyo et al.)

-Copies of NMR and HRMS spectra of CDg16 should be added to the Supplementary Information.

All the NMR and HRMS spectra of CDg16 were added as the Supplementary Method, Section 2 in 'Characterization of CDg16'.

Reviewer #2 (Remarks to the Author):

Park, SJ et al. "Imaging inflammation using an activated macrophage probe with Slc18b1 as the selective gating target" In this manuscript, the authors describe a library selection of a new probe based on an acridine orange parent structure for activated macrophages. Furthermore, the authors performed a biological selection to determine which transporter is responsible for the lead compound's import into activated macrophages. This is a well written manuscript with just a few grammatical errors. The search for a specific M1 polarization probe is important and a worthwhile goal. The authors performed in vitro and ex vivo fluorescence imaging in cultured cells and excised tissues of mice with limited correlative histologic confirmation of cell type or phenotype. This flaw and others listed below reduce the quality of this manuscript:

We thank the reviewer for the supportive comments and all the pointed experiments have been performed and summarized as below.

Q1. No ClogD or PSA values for acridine orange or for CDg16 are presented for comparison. These are crucial for predicting and explaining probe biodistribution in fatty tissues and in macrophages accumulating lipid as well as in systemic clearance and CNS uptake.

Thank you for the reviewer's comments. We calculated ClogD and tPSA values (by Chemicalize program from ChemAxon) and found the ClogD values of acridine orange and CDg16 were similar (2.93 & 3.31 at pH 7.4 and 2.01 & 1.78 at pH 4.5, respectively). Interestingly, the tPSA value was much lower in acridine orange compared to CDg16 (19.4 versus 114.4) (Supplementary Figure 6a), suggesting that CDg16 may be less (passively) permeable to the cells rather than acridine orange, but may show better selectivity through specific transport mechanism by Slc18b1.

Q2. Acridine orange does not seem to have been used as a comparative control. It would appear, for example, that CDg16 simply accumulates in cells with high densities of acidic or lipid dense lysosomes (like A.O.). Epithelial cancer cells were not included in uptake studies and these contain many similar vesicles.

We appreciated the reviewer's comments and we performed the suggested experiments. When we compared the staining pattern of acridine orange and CDg16, acridine orange could not discriminate M1 against non-activated Raw264.7 cells, and mainly stained nucleus both in non-activated and activated Raw264.7 cells (Supplementary Figure 6b).

In addition, we tested CDg16 in different types of epithelial cell lines derived from lung cancer (A549, H23, HOP62) and colon cancer (HT29, HCT15, SW620). None of the six cell lines showed detectable vesicle staining of CDg16 (Supplementary Figure 13).

Q3. This reviewer did not see any attempt to perform blocking studies to ascertain specificity for cells. In the ApoE^{-/-} model, arteries contain plaques rich in lipid pools, which may non-specifically solubilize this particular probe and if so, probe would be extracted out during routine paraffin embedded histology methods. The photos provided are macroscopic and would not differentiate between fat solubilization and cell uptake. The use of unfixed, frozen sections would be very helpful in delineating total probe distribution.

Thanks for the reviewer's comment, and we realized our method description was not clear enough. In all the results in this paper, we indeed used freezing tissue samples for epifluorescence imaging and immunohistochemistry, not paraffin embedded method. To clarify our experimental procedures, we revised the manuscript with more information in the Line 403-409 as follow:

“The aorta and paw samples were enucleated and immediately frozen for the cryosections. The samples were sectioned by the cryostat (Leica CM1950) with 10 μm thickness and mounted on the poly-L-lysine coated slides. The sectioned samples were fixed in 4% paraformaldehyde (PFA) for 15 minutes for immunohistochemistry. And the cell culture samples were also fixed in 4% PFA for 15 minutes for immunocytochemistry. After washing the sectioned and cell culture samples with PBS, the samples were treated with 1% bovine serum albumin (30min) for removing non-specific binding.”.

Moreover, we added new flow cytometry data using the dissociated aorta cells isolated from ApoE^{-/-} mouse to confirm the intracellular staining of CDg16 in the live cell status (Supplementary Figure 12).

Q4. Kupffer cells in liver (and macrophages in lung) in an ApoE^{-/-} mouse would be highly inflamed and no data were presented in these tissues as the identity of the livers and lungs in the supplementary data is unclear (control or inflamed?). Liver is dark and may easily quench visible green fluorescence, limiting the applicability of this probe.

We appreciate the important point out. Under the fluorescence stereomicroscope, we imaged not only the aortas (Figure 2 and Supplementary Figure 8), but also other organs using the same optical conditions (such as the exposure time and the binning) from the four groups (non-injected or CDg16-injected wild-type and ApoE^{-/-}) as shown in Supplementary Figure 9. Under the same imaging condition, CDg16 signal was not strong in livers under the stereomicroscope, may be due to the low level of liver inflammation in this model. To rule out the quenching effect of CDg16 in liver, we tested CDg16 in hepatitis model mice and confirmed a strong CDg16 signal, well overlapped with CD86 antibody. This results clearly show that CDg16 works well in the liver disease model (Supplementary Figure 14).

Q5. Please change your designation from “activated macrophages” to “reactive M1 phenotype macrophages”, if this is what you mean.

Thank you for the Reviewer's comments. We described M1 or M2 activated macrophages

instead of activated macrophages in manuscript and supplementary information.

Q6. M2 macrophages are also reactive and have just as many acidic lysosomes and should also express the Slc18b1 transporter as many tissue types do. It stands to reason that M1 probe trapping may be related to ROS ionization, rather than lysosome sequestration. The anthracene ring should be quite susceptible if not the oxazole. No experiments were presented showing M2 polarized macrophages and probe uptake. Histologic confirmation was so limited, readers cannot conclude whether M1 specificity is, in fact, a robust phenomenon. CD86 is expressed by several antigen presenting cell types. No additional markers were used of any kind such as CD68, CD206, CD163 or even iNOS. With fluorescence imaging, multi-channel marker use is encouraged for clarity.

We appreciated the reviewer's suggestion and we performed the experiment. THP-1 monocytes were used to polarize the cells into M1 and M2 macrophages, each of which was evaluated by its CD86 and CD206 expression through RT-PCR and immunofluorescence imaging (Supplementary Figure 11a, c). Interestingly, we found CDg16 can label M2 macrophages as well as M1 cells (Supplementary Figure 11b, c). As the reviewer pointed, M1 and M2 have different characteristics like ROS generation and ROS response may not be the reasonable mechanism for CDg16. Instead, we analyzed the expression level of SLC18B1 in the M2 macrophages and compared with M1. SLC18B1 expression was enhanced in both types of activated macrophages (M1 and M2 macrophages) in similar level (Supplementary Figure 21).

We also carefully examined the staining of CDg16 to the M1 and M2 macrophages in ApoE^{-/-} aorta tissues utilizing more CD markers as suggested and analyzed the result by flow cytometry analysis. Like the immunohistochemistry result, the population of CD45⁺CD38⁺ (41.6% of CD45⁺ cells) or CD45⁺CD86⁺ M1 cells (47.7% of CD45⁺ cells) were clearly stained with CDg16, and the small number of CD45⁺CD206⁺ M2 macrophages (2.1% of CD45⁺ cells) were also labeled with CDg16 (Supplementary Figure 12). Therefore, these results collectively suggest that CDg16 stained both M1 and M2 activated macrophages either *in vitro* and *in vivo* condition. So, we may keep the target of CDg16 as originally designated as activated macrophages.

While the search for an M1 specific probe continues, it is the opinion of this reviewer that the work presented in this manuscript lacks the experimental rigor needed to support the conclusions.

While the distinguishment of M1 and M2 is another interesting topic, understanding the common part of the development is also important. As a result of our work, with the reviewer's suggestion, we found the common basis of M1 and M2 in terms of the common target of CDg16, as of SLC18B1 expression. The discovered facts and discussions are

described in the later part of the paper (Line 146-166, 218-229).

REVIEWERS' COMMENTS:

Reviewer #1 (Remarks to the Author):

The revised manuscript by Chang et al. addresses all the concerns raised by the reviewers. Among many additional experiments, additional characterization including well-established markers of M1 and M2 macrophages has been provided and also the authors examined the correspondence between Slc18b1 and CD16g staining in macrophages and not only in HeLa cells.

As a minor point, it may be worth adding a recent new reference from the Ravichandran lab (Nature 2018, 714) on the characterization of SLC transporters in phagocytes. Altogether, I believe that this is a very round-up work and I can recommend the manuscript for publication in Nature Communications.

Response to Reviewers

Reviewers' comments:

Reviewer #1 (Remarks to the Author):

The revised manuscript by Chang et al. addresses all the concerns raised by the reviewers. Among many additional experiments, additional characterization including well-established markers of M1 and M2 macrophages has been provided and also the authors examined the correspondence between Slc18b1 and CD16g staining in macrophages and not only in HeLa cells.

As a minor point, it may be worth adding a recent new reference from the Ravichandran lab (Nature 2018, 714) on the characterization of SLC transporters in phagocytes. Altogether, I believe that this is a very round-up work and I can recommend the manuscript for publication in Nature Communications.

We sincerely appreciate your positive reviews and the valuable comment.

We applied the update of SLC transporter in page 9, line 196, and added the new reference in the revised manuscript (Reference 23).